# Understanding Adaptive, Multiscale Temporal Integration In Deep Speech Recognition Systems

**Menoua Keshishian**[*]
Department of Electrical Engineering
Zuckerman Mind Brain Behavior Institute
Columbia University
New York, NY 10027
mk4011@columbia.edu

**Sam V. Norman-Haignere**[*]
Department of Electrical Engineering
Zuckerman Mind Brain Behavior Institute
Columbia University
New York, NY 10027
sn2776@columbia.edu

**Nima Mesgarani**
Department of Electrical Engineering
Zuckerman Mind Brain Behavior Institute
Columbia University
New York, NY 10027
nima@ee.columbia.edu

## Abstract

Natural signals such as speech are hierarchically structured across many different timescales, spanning tens (e.g., phonemes) to hundreds (e.g., words) of milliseconds, each of which is highly variable and context-dependent. While deep neural networks (DNNs) excel at recognizing complex patterns from natural signals, relatively little is known about how DNNs flexibly integrate across multiple timescales. Here, we show how a recently developed method for studying temporal integration in biological neural systems – the temporal context invariance (TCI) paradigm – can be used to understand temporal integration in DNNs. The method is simple: we measure responses to a large number of stimulus segments presented in two different contexts and estimate the smallest segment duration needed to achieve a context invariant response. We applied our method to understand how the popular DeepSpeech2 model learns to integrate across time in speech. We find that nearly all of the model units, even in recurrent layers, have a compact integration window within which stimuli substantially alter the response and outside of which stimuli have little effect. We show that training causes these integration windows to shrink at early layers and expand at higher layers, creating a hierarchy of integration windows across the network. Moreover, by measuring integration windows for time-stretched/compressed speech, we reveal a transition point, midway through the trained network, where integration windows become yoked to the duration of stimulus structures (e.g., phonemes or words) rather than absolute time. Similar phenomena were observed in a purely recurrent and purely convolutional network although structure-yoked integration was more prominent in the recurrent network. These findings suggest that deep speech recognition systems use a common motif to encode the hierarchical structure of speech: integrating across short, time-yoked windows at early layers and long, structure-yoked windows at later layers. Our method provides a straightforward and general-purpose toolkit[2] for understanding temporal integration in black-box machine learning models.

---

[*] These authors contributed equally.
[2] Code available at: https://github.com/naplab/PyTCI

# 1 Introduction

A central challenge of representing natural signals, such as speech and music, is that they are structured across many different timescales (Chomsky and Halle, 1968; Lerdahl and Jackendoff, 1985; Hickok and Poeppel, 2007). Speech, for instance, is composed of phonemes, syllables, and words spanning tens to hundreds of milliseconds, and the duration of these structures varies widely due to a myriad of factors (e.g., talker, speaking rate, prosody, etc.) (Figure 1). Intelligent systems, whether biological or artificial, must learn to flexibly integrate across these different timescales to derive meaning from natural signals. DNNs have proven remarkably effective at learning complex temporal structures from natural signals, such as speech, and the representations learned by these networks show non-trivial similarities to those present in the brain (Kell et al., 2018). However, relatively little is known about how deep speech recognition systems flexibly integrate across multiple timescales. Here, we address this gap by using a recently developed method for studying temporal integration in biological neural systems – the temporal context invariance (TCI) paradigm (Norman-Haignere et al., 2020) – to probe how a widely used deep speech recognition system (DeepSpeech2) learns to integrate across multiple timescales.

We study temporal integration by measuring how training alters the integration window of DNN units. Integration windows are defined as the time window when stimuli alter the neural response and outside of which stimuli have little effect (Theunissen and Miller, 1995; Hasson et al., 2008). Because this definition is simple and general, integration windows can be used to measure and compare the analysis timescale of responses from different models, stimuli, and modalities. While convolutional neural networks (CNNs) have a well-defined upper bound on their integration window, determined by the size of the temporal kernels, the effective integration window, when stimuli empirically alter the response, can differ substantially from this upper bound (Luo et al., 2016) and change substantially with training, as we show. Recurrent neural networks (RNNs) can in principle integrate over arbitrarily long time windows (Hochreiter and Schmidhuber, 1997; Sussillo and Barak, 2013), but in practice, RNNs also show time-limited effective integration windows within which stimuli have a much greater impact on the measured response (Mahto et al., 2020; Chien and Honey, 2020).

We have recently designed a method to estimate the effective integration window of an arbitrary response by presenting segments of natural stimuli (here speech) in two different random orders such that each segment occurs in two different contexts (i.e., is surrounded by different stimuli) (Figure 2a) (Norman-Haignere et al., 2020). If the integration window is less than the segment duration, there will be a moment when the response is the same across the two contexts. We can thus vary the segment duration in order to find the smallest segment duration outside of which stimuli have little effect on the response. The random ordering of segments makes it possible to use a simple correlation measure to quantify at each moment in time the degree to which the response is context invariant (Figure 2b), as detailed below. The goal of this paper is to introduce this paradigm to the machine learning community and to use this paradigm to understand how deep speech recognition systems learn to flexibly integrate across multiple timescales in speech.

# 2 Relationship with prior work

Understanding latent representations of DNNs is a highly active area of research. A variety of feature visualization techniques have been developed to understand image-trained networks (Zeiler and Fergus, 2014; Mahendran and Vedaldi, 2015; Yosinski et al., 2015) with less work focused on understanding speech-trained networks (Huang et al., 2015; Krug and Stober, 2019; Beguš and Zhou, 2021; Keshishian et al., 2020). Many visualization methods are based upon calculating the gradient of the network's response with respect to the input (Springenberg et al., 2014; Bach et al., 2015; Selvaraju et al., 2016; Shrikumar et al., 2017; Sundararajan et al., 2017; Samek et al., 2021). While for a linear system, the temporal extent of the gradient is equivalent to the integration window, for a nonlinear system, the link between the local gradient and the overall computation is unclear (Balduzzi et al., 2017), and as a consequence, there is a need for empirical methods that can estimate the effective integration window of highly nonlinear systems (Norman-Haignere et al., 2020; Chien and Honey, 2020).

Probing tasks have been used to examine the representation of particular structures, such as phonemes, across different layers of speech recognition networks (Belinkov and Glass, 2017; Nagamine et al., 2015; Nagamine and Mesgarani, 2017), and a recent study examined the geometry of how invariant

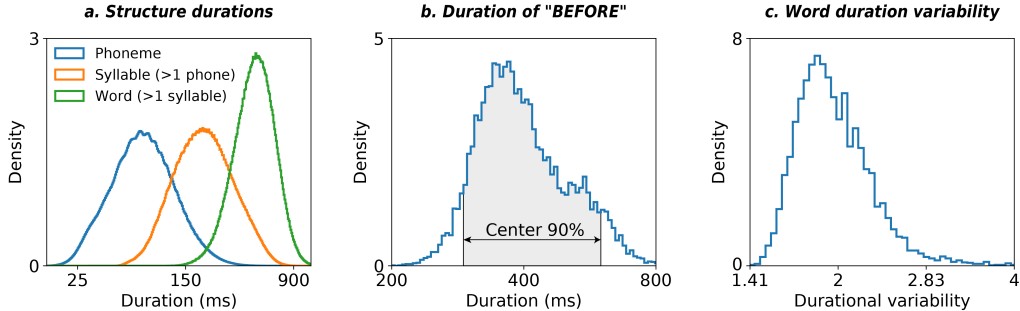

Figure 1: Duration statistics of speech structures in the LibriSpeech corpus. (a) Histogram of durations for all phonemes, multi-phoneme syllables, and multi-syllable words. (b) Histogram of durations for all utterances of the word "before" in the corpus. The central 90% interval is marked in gray with the 5th and 95th percentile indicated. (c) Histogram of durational variability across all words in the corpus. Durational variability is measured as the ratio between the 95th and 5th percentile duration for each word (as illustrated in panel b).

feature representations evolve over time (Stephenson et al., 2020). However, to our knowledge, little is known about how speech-trained networks learn to integrate across time, despite the centrality of temporal integration to the task of speech recognition.

Neuroscience has produced a variety of experimental methods for studying sensory timescales. Scrambling methods have been used to examine selectivity for intact temporal structure at different timescales (Hasson et al., 2008; Overath et al., 2015). Chien and Honey (2020) recently introduced an experimental method that estimates the delay needed for a response to become context-invariant after a stimulus change, and they used this method to analyze neural language models (Chien and Honey, 2020). However, this method does not directly measure the time window in the stimulus that is needed to achieve an invariant response, which requires measuring responses to segments of differing durations. Moreover, speech recognition presents many unique challenges that are not present in language modeling. For example, a key contribution of our paper is showing that there is a transition from fixed, time-yoked integration to adaptive, structure-yoked integration in deep speech recognition models, a distinction that is not meaningful in language models where the input is inherently structure-based (i.e., words).

## 3 Temporal context invariance (TCI) paradigm

### 3.1 Stimuli

The input to our analysis is a set of sound segments of varying duration. We tested durations ranging from 20 milliseconds and 2.48 seconds in pseudo-logarithmic steps (20, 40, 60, 80, 100, 120, 140, 160, 200, 240, 280, 340, 400, 480, 580, 700, 840, 1000, 1200, 1440, 1720, 2060, 2480 ms).

Sound segments were excerpted from the LibriSpeech corpus' *dev-clean* and *test-clean* sets. We attempted to choose segments that were composed of words spoken at a moderate rate and which also had a moderate amount of natural variation in their duration, such that stretching or compressing the word would not place the word outside of its typical duration (stretching and compression analyses described below). To this end, we first selected all words spoken at least 100 times in the corpus. For each word, we computed a histogram of durations, and we selected all words where the difference between the 5th and 95th percentile duration was at least a factor of 2 (see Figure 1c for a histogram of this durational variability metric across words). For each word, we then randomly selected up to 100 of its utterances that fell within the central 20% of its duration distribution. This resulted in a pool of 6,184 utterances across 1,095 unique words. Finally, for each utterance, we excerpted 23 segments, one per duration, with the middle of the word at the center of the segment (if the segment duration was shorter than the word, we excerpted just the middle portion of the word). This selection procedure resulted in 6,184 unique segments per segment duration. We note, that while we used a speech-specific recipe to extract segments, having in mind the types of experiments we would

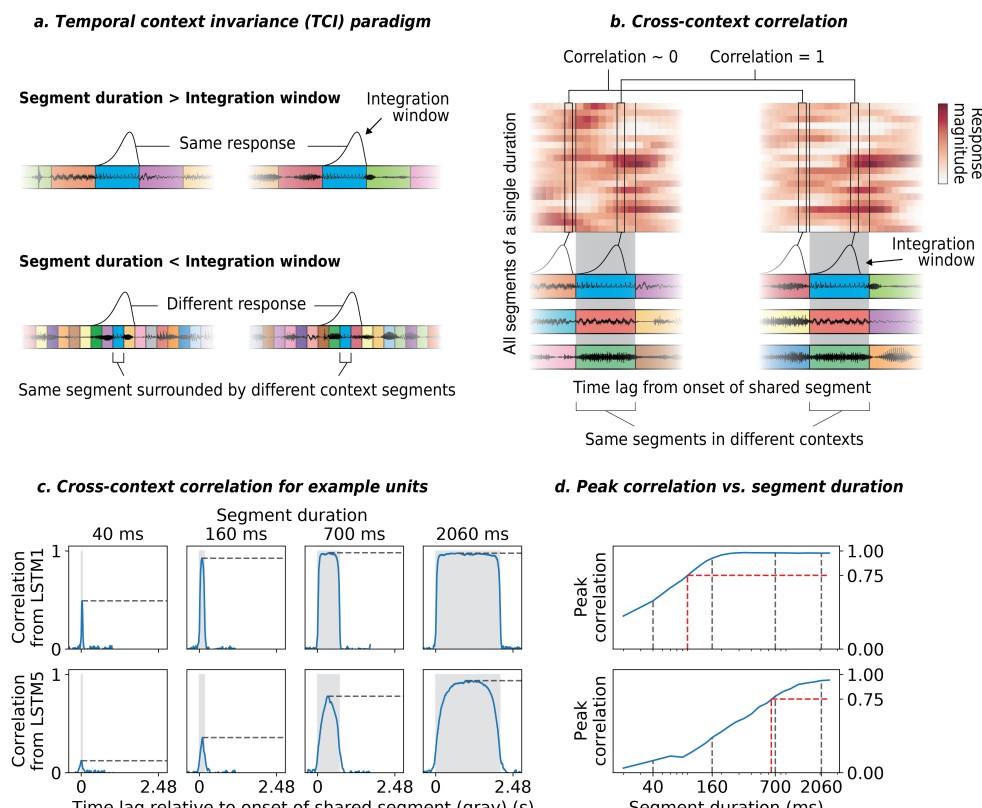

Figure 2: Temporal context invariance (TCI) paradigm. (a) Schematic of the paradigm used to measure integration windows. Segments of natural stimuli (here speech) are presented using two different random sequences. As a consequence, each segment is surrounded by a different context in each sequence. If the segment duration is longer than the integration window, there will be a moment when the integration window is fully contained within the segment, yielding a similar response across the two contexts. (b) Schematic of the analysis used to assess context invariance for a single unit and segment duration. We organize the responses of a unit to all segments of a given duration as a [segment x time] matrix (the segment-aligned response or SAR matrix). Each row contains the unit's response to a single segment, aligned to segment onset. A separate SAR matrix is calculated for each sequence, such that corresponding rows contain the response to the same central segment, surrounded by different context segments. We correlate corresponding columns across the two SAR matrices (the cross-context correlation), schematized by linked columnar boxes. When the integration window overlaps the context segments (first box pair), we expect a correlation of 0 since the context is random. If the integration is less than the segment duration, there will be a moment/lag when it is fully contained within the shared central segments (gray region), yielding a correlation of 1 at that moment (second box pair). (c) The cross-context correlation for two example units from different layers of DeepSpeech2 (LSTM1 and LSTM5). Each plot shows a different segment duration. The gray region shows the time interval when the shared segment was present. Black dashed lines show the peak of the cross-context correlation across all lags. (d) The peak correlation for all tested segment durations. Black dashed lines highlight the segment durations shown in panel c. The red dashed line indicates the segment duration needed to achieve a peak correlation value of 0.75, which is how integration windows were defined.

conduct, this is not necessary or intrinsic to the TCI method. The segment extraction recipe can change depending on the nature of the stimuli and the intended experiments. Time alignments for phonemes and words of the LibriSpeech dataset were extracted from (Lugosch et al., 2019), which used the Montreal Forced Aligner to align the transcripts to sounds (McAuliffe et al., 2017). Syllables were identified based on the extracted phonemes using the *syllabify* python library (Gorman, 2013).

All of the segments from a given duration were randomly ordered and then concatenated with cross-fading at the boundary to minimize transition artifacts (20ms half-Hanning window). We used two different random orderings of the segments such that each segment occurred in two different contexts (Fig 2a). Due to limited memory, we divided the randomly ordered segments into 48-second stimuli with 16-seconds of overlap between consecutive stimuli. The responses to the first and last 8 seconds of each stimulus were discarded to avoid padding artifacts due to batching.

## 3.2 DeepSpeech2 model

We applied our TCI paradigm to measure integration windows from a DeepSpeech2 model (Amodei et al., 2016) (trained for 20 epochs). We used a publicly available implementation of this model (Naren, 2019). Our DeepSpeech2 model consists of two 2D convolutional layers (Conv1/2), five bi-LSTM layers (LSTM1/2/3/4/5), and a softmax readout that is trained to predict graphemes using a CTC loss (Graves et al., 2006). All layers except the readout included batch normalization. The model reached a word error rate (WER) of 7.6% on the LibriSpeech *test-clean* set after training for 20 epochs. We analyzed logarithmically transformed probabilities for the softmax readout (to avoid the high sparsity of probabilities). We compared results from our trained model with a randomly initialized model to assess effects of training.

All models were implemented in PyTorch (Paszke et al., 2019) and trained using PyTorch Lightning (Falcon, 2019) on the training set of the LibriSpeech corpus (Panayotov et al., 2015). We used the CTC loss (Graves et al., 2006), the Adam optimizer (learning rate: $1.5e-4$, weight decay: $1e-5$) (Kingma and Ba, 2014), and a batch size of 64. To improve model generalization, we augmented the stimuli by adding background noise, reverberation, and frequency masking. Augmentations were performed using the Sound eXchange (SoX) backend of the audio library for PyTorch (*torchaudio*).

Training and inference of all models were performed on NVIDIA A40 GPUs (one per training/inference) at the internal cluster at the Zuckerman Institute of Columbia University. Each epoch of training took approximately 2.5 hours for each model, for a total of 110 epochs across all models. For each experiment, we ran inference across three time-stretches and 23 segment durations, for an average of 22 hours per experiment. Inference was substantially faster for the convolutional model.

## 3.3 Cross-context correlation

Our analysis involves correlating the response of a unit to different stimulus segments across two different contexts, an analysis we refer to as the "cross-context correlation" (Figure 2b). The analysis is applied separately to different segment durations. For each segment duration, we have two stimulus sequences ($x_A[t]$ and $x_B[t]$). The stimulus sequences could be vector-valued as in a spectrogram or scalar as in a waveform. Each sequence contains all the segments from that duration, randomly ordered and concatenated. For simplicity, we will ignore the fact that we divided our sequences into 48-second stimuli due to memory limitations and assume that there are just two long sequences, each of which creates a unique context for each segment. Each unit in our model produces an output sequence in response to these two stimuli (we do not specify the index of the unit to simplify notation):

$$r_A[t] = f\left(x_A[t]\right) \tag{1}$$
$$r_B[t] = f\left(x_B[t]\right) \tag{2}$$

where $f(\cdot)$ is a scalar function specifying the stimulus-response mapping, here a deep network. From these two response sequences we create two segment-by-time matrices, which we refer to as the segment-aligned response (SAR) matrices (see Figure 2b for a schematic):

$$\text{SAR}_A[s,t] = r_A\left[t - o_A[s]\right] \tag{3}$$
$$\text{SAR}_B[s,t] = r_B\left[t - o_B[s]\right] \tag{4}$$

where $o_A[s]$ is the onset in samples of segment $s$ in sequence $A$, and $o_B[s]$ is the onset of that same segment in sequence $B$. Thus, each row of the SAR matrix contains the response sequence that surrounds a particular stimulus segment. Corresponding rows contain the response sequence to the same segment but surrounded by different context segments. The cross-context correlation ($\rho_{ccc}[t]$) is computed by correlating corresponding columns across the two SAR matrices:

$$\rho_{ccc}[t] = \mathrm{corr}(\mathrm{SAR}_A[:,t], \mathrm{SAR}_B[:,t]) \tag{5}$$

where $\mathrm{corr}$ indicates the Pearson correlation (though we see no reason why another measure of dependence could not be used). The set of time lags ($t$) relative to segment onset should be larger than the integration windows being measured. Here we used time lags from $-1$ seconds to $T + 1$ seconds, where $T$ is the segment duration. The sampling rate was 100 Hz.

The goal of our analysis is to assess if there is a lag when the response is the same across contexts. Before segment onset (lag<0), the cross-context correlation should be near zero for a causal response, since the integration window must overlap the preceding segments, which are random across contexts. As time progresses, the integration window will start to overlap the shared segment, and the cross-context correlation should increase. Critically, if the integration window is less than the segment duration, there will be a lag where the integration window is fully contained within the shared segment, and the response will thus be the same across contexts, yielding a correlation of 1. For a non-causal response, such as a bidirectional LSTM, we expect the same pattern, but the build-up of the cross-context correlation can begin before the onset of the shared segment.

We plot the cross-context correlation for segments of increasing duration for two example units from different layers of the DeepSpeech2 network (LSTM1 and LSTM5) (Figure 2c). For both units, the cross-context correlation started at zero, rose near the onset of the shared segment, and fell back to zero after the offset of the shared segment. Critically, the peak of the cross-context correlation varied across segment durations and units. For the unit from the LSTM1 layer (top panel), the peak of the cross-context correlation was close to 1 for segment durations as short as 160 milliseconds. For longer segment durations, the cross-context correlation remained high for an extended period, indicating that the unit's integration window was short relative to the segment duration and thus remained within the shared segments for an extended period. By comparison, the unit from the LSTM5 layer required much longer segment durations for the cross-context correlation to approach 1, and the build-up/fall-off of the cross-context correlation with lag was slower, implying a substantially longer integration window. These general trends were observed in virtually all units (Appendix Figure 1 shows several more units).

We quantified the integration window of each unit by measuring the peak of the cross-context correlation across lag as a function of the segment duration (Figure 2d). The units showed a clear pattern where the peak correlation rose steeply as the segment duration was increased, and then gradually approached an asymptotic value. This asymptotic value was high for nearly all units (Appendix Figure 2; 98.9% of units had an asymptotic correlation greater than 0.9), indicating a mostly context-invariant response. We defined the integration window of each unit as the segment duration needed to achieve a correlation value of 0.75 (estimated by linearly interpolating the correlation vs. segment duration curves shown in Figure 2d). Results were robust to the choice of threshold, though higher thresholds inevitably result in longer integration windows.

## 4 Effect of training on network integration windows

Figure 3 plots the distribution of integration windows for the units in each layer for both trained and randomly initialized models. The integration windows from the random model varied little with layer, particularly amongst the five LSTM layers (median integration windows varied between 167 and 204 ms across the LSTM layers). We found that training caused the integration windows of the early layers (Conv1 - LSTM2) to shrink and the integration windows of later layers (LSTM3 - softmax) to expand (Wilcoxon rank-sum test comparing trained and random models across units, $p < 1\mathrm{e}{-5}$). For the trained model, there was an approximately 10-fold increase between Conv1 (median window: 58 ms) and LSTM5 (median window: 592 ms). This pattern appears to qualitatively mimic the hierarchical nature of temporal integration in biological auditory systems (Khatami and Escabí, 2020; Norman-Haignere et al., 2020).

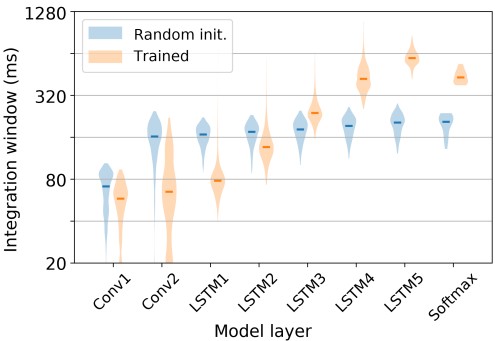

Figure 3: Effect of training on network integration windows. Violin plots show the distribution of integration windows across all units from each layer for a trained (orange) and randomly initialized model (blue). The violin plots show the center 99% of the distribution, discarding extreme outliers. Horizontal lines indicate distribution medians.

## 5 Time- vs. structure-yoked integration

The duration of speech structures such as phonemes and words is highly variable (Figure 1). Thus, a natural question is whether the time windows measured from the network reflect a static time window or instead reflect the duration of the structures or features to which a unit responds. A simple way to address this question is to time-compress or time-stretch the stimulus such that all stimulus structures get mapped to a new timescale. If the integration window is time-based, then the integration window should remain fixed, while if the integration window is structure-driven, it should appear to shift in proportion to the degree of stretching/compression.

Stretching and compression were implemented using the SoX backend of *torchaudio*, which uses the WSOLA algorithm to modify the rate of speech without changing the pitch. Stimuli were stretched or compressed by 20%. Results were robust to the amount of stretching/compression (see Appendix Figure 3) and the algorithm used to perform stretching/compression (see Appendix Figure 4). WER on LibriSpeech *test-clean* was best for natural speech (7.6%) and worst for compressed speech (10.4%), with stretched speech (8.3%) only slightly worse than natural speech.

Figures 4a–f show scatter plots of integration windows measured for stretched vs. natural speech for different model layers (each dot corresponds to one unit; note that for this analysis only we used a stretch factor of 60% so that it was visually easier to see the differences between stretched vs. natural speech). For early layers (e.g., Conv1), we found that integration windows were very similar for natural vs. stretched speech. In contrast, for late layers (e.g., LSTM4), we found that integration windows increased by a factor that was nearly the same as the amount of stretching in the stimulus, suggesting integration windows were determined nearly fully by the duration of stimulus structures.

To quantify this change, we computed the following index, reflecting the degree to which the integration window adapts to the duration of stimulus structures:

$$\text{Adaptation index} = \frac{I_{mod}/I_{ref} - 1}{D_{mod}/D_{ref} - 1} \tag{6}$$

where $I_{mod}$ is the measured integration window for stretched/compressed speech and $I_{ref}$ is the integration window for natural speech. $D_{mod}$ and $D_{ref}$ are the relative durations of the stimulus structures for the corresponding conditions (i.e., $D_{mod}/D_{ref}$ would be 1.2 for a stretch factor of 20%). An index of 0 indicates a fixed, time-yoked window, while a value of 1 indicates a fully adaptive, structure-yoked window. Figures 4g–h plot this adaptation index across all layers for stretched (top panel) and compressed speech (bottom panel) from both trained (orange violins) and randomly initialized (blue violins) networks.

We found that time-stretching/compression had little impact on measured integration windows from an untrained model, yielding adaptation indices near 0. In contrast, for the trained model, we found there was a transition point between layers LSTM1 and LSTM3, where integration windows shifted from mostly time-yoked integration (median adaptation index of 0.19/0.21 in LSTM1 for stretched/compressed speech) to mostly structure-yoked integration (median adaptation index of

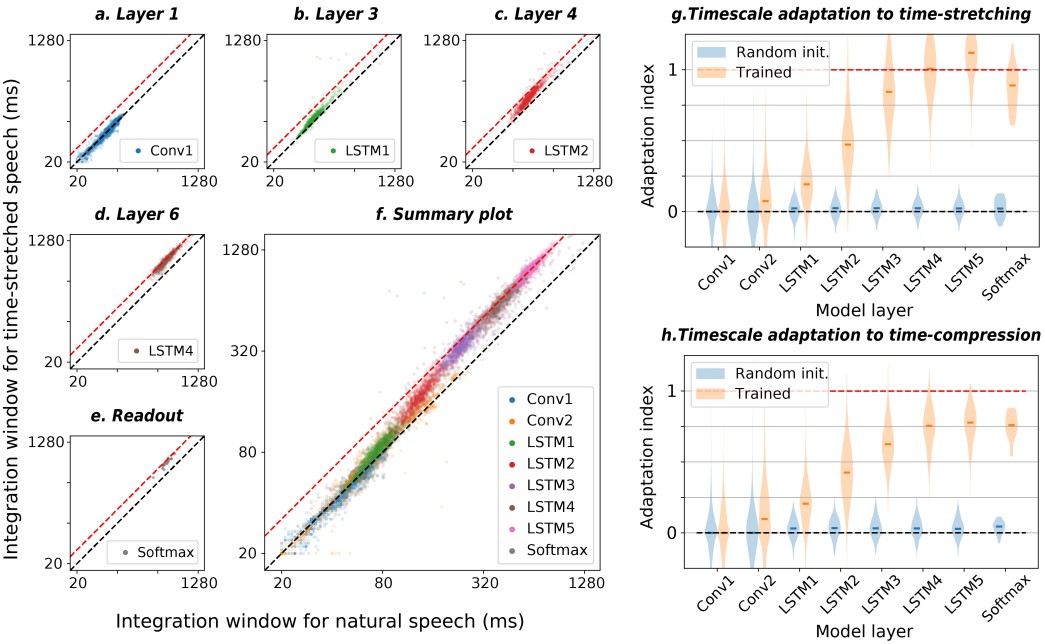

Figure 4: Integration windows for time-stretched/compressed speech. (a–e) For each layer, a scatter plot shows the integration windows measured for stretched vs. natural speech. Each dot corresponds to a single unit. The black dashed line is the line of unity, showing what one would expect if integration windows were unaffected by stretching. The red dashed line shows what one would expect if integration windows expanded in direct proportion to the magnitude of stretching. (f) Integration windows from all layers are jointly plotted on the same graph (colored by layer), illustrating the progression from short, time-yoked windows to long, structure-yoked windows. (g-h) For each unit, we computed an adaptation index, reflecting the degree to which its integration window shifted with the magnitude of stretching/compression. A value of 0 indicates a fixed, time-yoked window (i.e., near the black dashed line in panels a-f), while a value of 1 indicates a fully adaptive, structure-yoked window (i.e., near the red dashed line in panels a-f). Violin plots show the distribution of adaptation indices across all units from each layer. Results are plotted for trained (orange) and randomly initialized models (blue), measured with either stretching (g) or compression (h). Violin plots show the center 99% of the distribution, discarding extreme outliers. Horizontal lines indicate distribution medians.

0.84/0.63 in LSTM3 for stretched/compressed speech). The magnitude of adaptation was slightly larger for stretched vs. compressed speech in the later layers (Wilcoxon rank-sum test across units: $p < 1\mathrm{e}{-5}$ for LSTM2 through LSTM5). This observation suggests that the network was better able to adapt its integration window for stretched speech, which might be related to the network's better performance for stretched (8.3% WER) vs. compressed speech (10.4% WER). Collectively, these findings suggest that the network learns to integrate across short, fixed time windows at early layers and long, structure-yoked windows at later layers.

# 6 Architecture comparison

Theoretical arguments suggest that gated recurrent architectures, such as LSTMs, may be particularly well-suited to adaptive integration (Tallec and Ollivier, 2018). Thus, a natural question is whether our findings depend upon the presence of gated RNNs. To address this question, we analyzed two additional models: a fully-recurrent model with 7 bi-LSTM layers; and a fully-convolutional model based on QuartzNet, which uses separable convolutions and residual connections (Kriman et al., 2020) (simply repeating the convolutional layers in DeepSpeech2 produced an architecture with very poor performance). We simplified the QuartzNet model by taking the 5x5 model (see (Kriman et al., 2020)), removing the fully-connected $C_3$ layer, changing the number of output channels ($C$) of all hidden layers to 1024, changing the kernel size ($K$) of all hidden layers to 41, and using

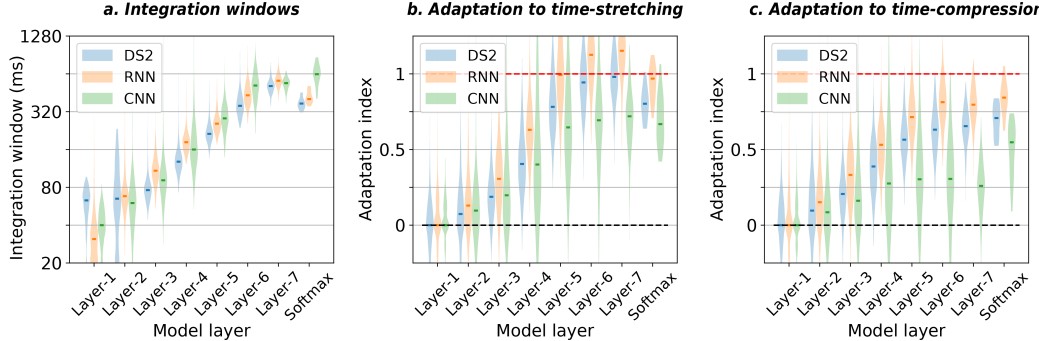

Figure 5: Effect of model architecture. (a–c) Integration window and adaptation index distributions for DeepSpeech2 (blue), fully-recurrent (orange), and fully-convolutional (green) architectures. The violin plots show the center 99% of the distribution, discarding extreme outliers. Horizontal lines indicate distribution medians.

symmetrical padding such that the model would be non-causal, similar to the bi-LSTM modules. For the CNN model, we do not show results from the intermediate layers within each residual block to make the results easier to compare with the other models (see Appendix Figure 5 for all layers). We approximately matched the performance of the three models (DeepSpeech, RNN, CNN) by selecting the training epoch that led to a WER closest to 10% on the LibriSpeech *test-clean* set (10 epochs for DeepSpeech2 and RNN, 50 epochs for the CNN).

Both the convolutional and recurrent networks learned a similar hierarchy of integration windows for natural speech, with substantially longer integration windows in later network layers (Figure 5). Moreover, both convolutional and recurrent models showed greater adaptation at later layers, similar to DeepSpeech2. However, the magnitude of adaptation was greater for both DeepSpeech2 and the fully recurrent model, compared with the convolutional model (Figure 5b-c). These results indicate that our core findings do not depend on recurrence, but also suggest that RNNs may show a greater predisposition for adaptive integration.

## 7 Robustness to task, stimuli, and context

We performed several additional experiments to test the robustness of our findings. We found similar results for a model directly trained to recognize words, as opposed to graphemes (Appendix Figure 6). We also observed similar results for frequent and infrequent words, demonstrating that temporal adaptation is not limited to frequent words. Our paradigm also allows us to consider "natural contexts", when a segment is a subset of a longer segment and thus surrounded by its actual context in natural speech, in addition to the case described so far when a segment is surrounded by random other segments. While one of the two contexts being compared has to be random so that the contexts differ, the other context can be random or natural (see Appendix Figure 7 for a schematic). In practice, we observed similar results using random-random and random-natural contexts (Appendix Figure 7), suggesting that our findings are robust to the particular type of context used to assess invariance.

## 8 Conclusion

We have described an approach for understanding how DNNs learn to integrate across time in complex natural signals, inspired by recent work in understanding temporal integration in biological systems (Norman-Haignere et al., 2020). Our method involves presenting segments of natural stimuli in two different contexts and estimating the smallest segment duration outside of which responses are largely invariant to the surrounding context stimuli. Using this approach, we have shown that deep speech recognition models integrate across time using short, fixed windows at early layers and long, adaptive windows at later layers. These findings were robust across the architecture (Figure 5), task (Appendix Figure 6), speech stimuli (Appendix Figure 8), and the type of context used to assess context invariance (Appendix Figure 7).

We emphasize that our core method is simple and general. Nothing about our approach is specific to speech or the particular models tested. We thus believe that the approach described in this paper should be widely applicable to understanding how intelligent systems integrate across the complex temporal structures that compose natural signals.

# 9 Limitations

The uniform stretching/compression used to assess adaptation is unnatural and thus it is possible that we might observe greater adaptation with more natural forms of stretching/compression. Uniform stretching/compression is experimentally useful because it shifts the timescale of all stimulus structures, and thus should produce a shift in integration windows regardless of the particular structures a unit represents.

Due to time limitations, we were only able to explore a limited number of architectures, tasks, and stimuli, though we observed similar results across all of these manipulations. Future work could thus expand upon our findings by exploring a wider range of architectures (e.g., attention-based architectures), tasks (e.g., speaker recognition), and stimuli (e.g., music).

# 10 Broader impacts

Our findings raise questions about the extent to which biological and artificial systems use similar mechanisms to integrate across time. The hierarchy of temporal integration windows learned by DeepSpeech2 qualitatively mimics the hierarchy of integration windows observed in biological auditory systems (Khatami and Escabí, 2020; Norman-Haignere et al., 2020). A key long-term goal of this work is to build computational models that can mimic the mechanisms of temporal integration in the brain. Building accurate computational models of brain responses has many potential benefits, including better understanding and treating sensory deficits. We note that neural network training consumes large amounts of energy which could have a negative impact on the environment.

## Acknowledgments and Disclosure of Funding

This work was funded by NIH (1K99DC018051-01A1 to SNH, R01DC014279 and 1R01EB028155 to NM) and a grant from Marie-Josée and Henry R. Kravis.

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
