# Appendix

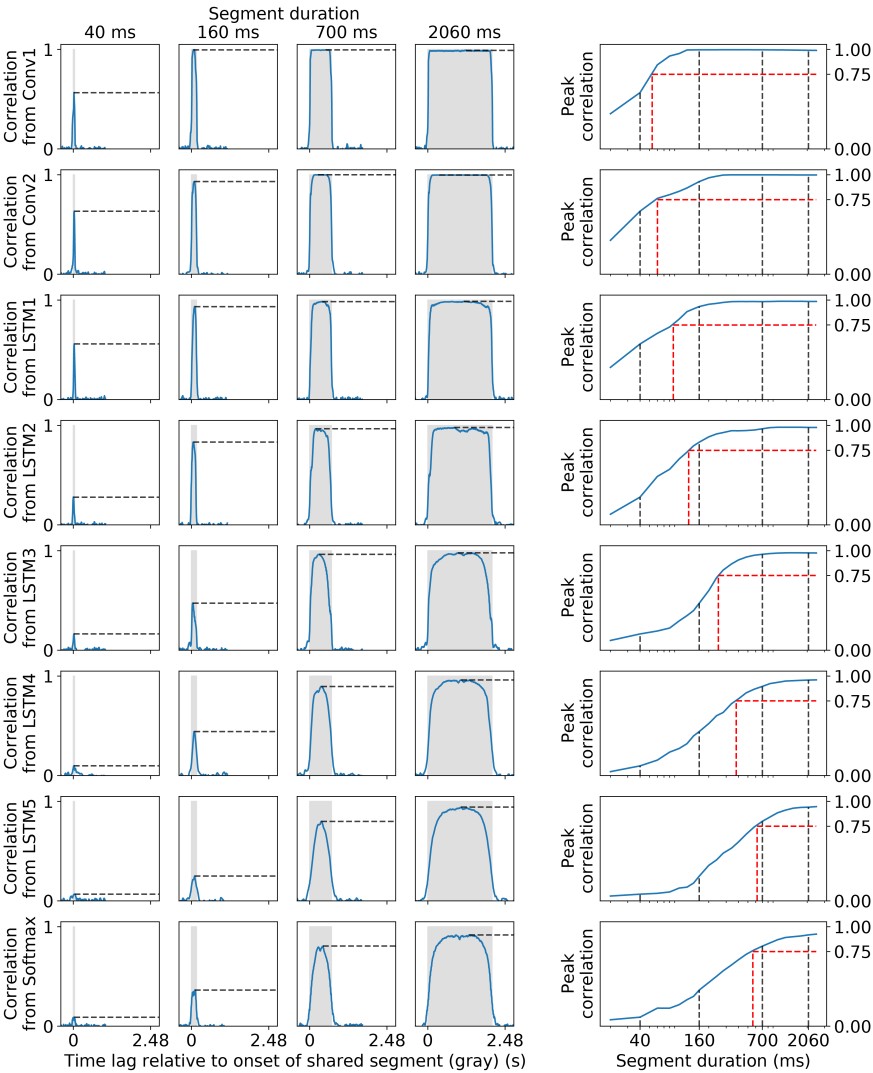

Appendix Figure 1: Cross-context correlation for additional units. This figure plots the cross-context correlation for an example unit from each of the eight layers. Format is the same as Figure 2c&d.

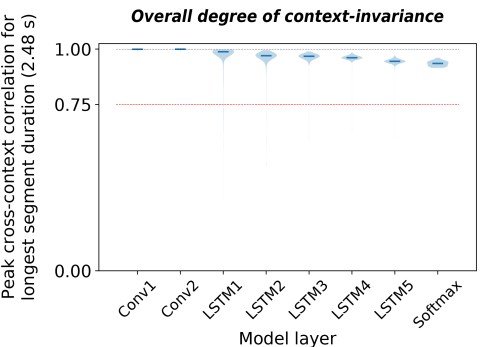

Appendix Figure 2: Overall degree of context invariance. The peak correlation vs. segment duration curve tended to approach an asymptotic value at long segment durations (see Figure 2d). For simplicity, we estimated this asymptotic value for each unit by measuring the peak cross-context correlation across lag for the longest segment duration tested (2.48 seconds) (i.e., the rightmost values in the curves shown in Figure 2d). This figure plots the distribution of this measure across the units from each layer. Convolutional layers have a maximum value of 1, as expected since they have a well-defined upper bound on their integration window. The LSTM layers also showed high maximum values (median correlation value across units was above 0.93 for all layers), indicating a mostly context-invariant response. The red dashed line indicates the 0.75 threshold used to define the integration window of each unit. Horizontal lines in the violin plots indicate distribution medians.

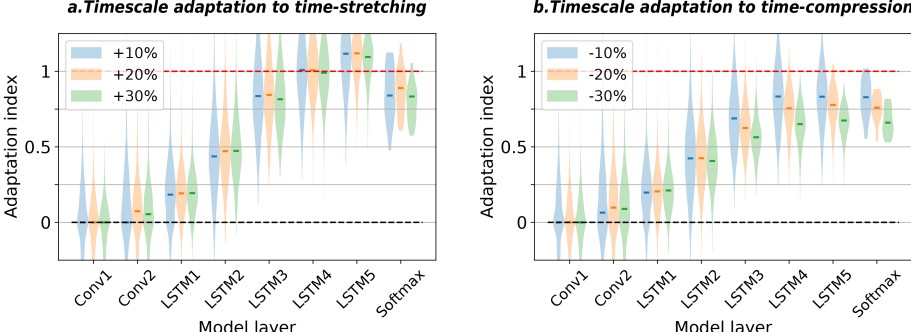

Appendix Figure 3: Effect of stretching/compression factor. (a–b) Adaptation indices from different layers, measured using different magnitudes of stretching (a) and compression (b). Results are similar across different stretching/compression magnitudes although there is a slight drop in the degree of adaptation for large compression magnitudes. Horizontal lines in the violin plots indicate distribution medians.

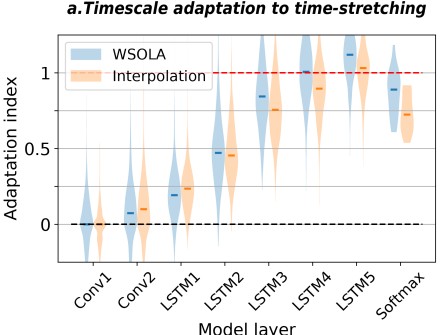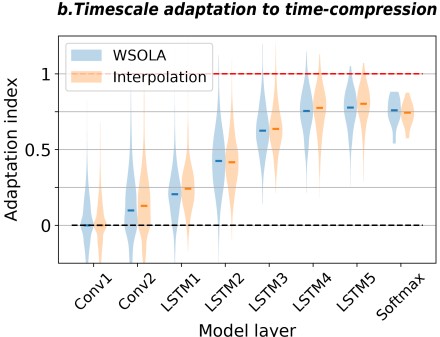

Appendix Figure 4: Effect of stretching/compression method. Speech was stretched/compressed using either waveform similarity overlap add (WSOLA as implemented by SoX, blue) or linear spectrogram interpolation (orange) (note that the input to DeepSpeech2 is a spectrogram). Adaptation indices are shown for stretched (a) and compressed speech (b) using each of the two methods. Both methods show similar trends. Horizontal lines in the violin plots indicate distribution medians.

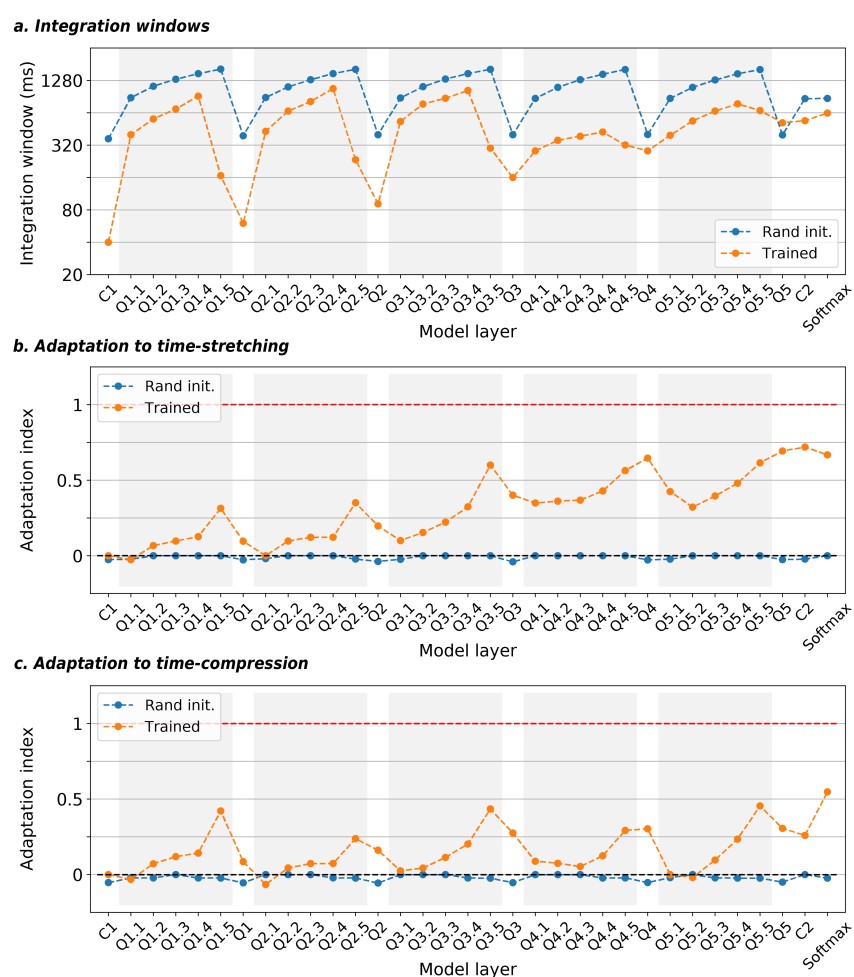

Appendix Figure 5: Results for all layers in the Quartznet-based CNN model. For simplicity, Figure ADD does not show intermediate layers within each residual block for the CNN model. Here we plot results from all layers, including layers within each residual block. (a) Integration windows measured from natural speech for the trained and randomly initialized model. Gray region highlights the layers not shown in Figure 5. (b-c) Adaptation indices for time-stretched (b) and compressed speech (c) for all model layers. $C_1$ and $C_2$ are convolutional blocks, $Q_1..Q_5$ are the output of the residual blocks, $Q_{1.1}...Q_{1.5}$ are the convolutional layers of $Q_1$, and so on.

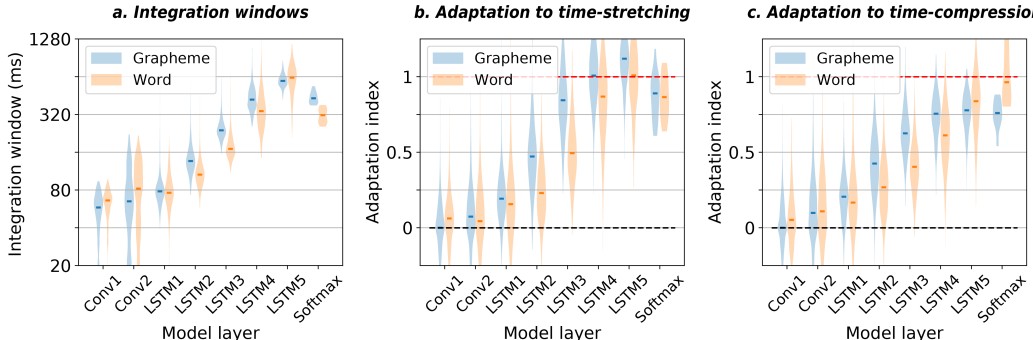

Appendix Figure 6: Results for DeepSpeech2 model trained to predict words instead of graphemes. (a) Integration windows measured using natural speech. This figure plots the distribution of integration windows from different layers of DeepSpeech2, trained to predict graphemes (blue, same as Main text) or words (orange) using a CTC loss. Same format as Figure 3. (b-c) Adaptation indices from different model layers, reflecting the degree to which the unit integration windows are yoked to structure vs. time. Results are plotted separately for time-stretched (b) and time-compressed speech (c) for models trained on graphemes (blue, same as Main text) or words (orange). Same format as Figure 4. Trends are qualitatively similar between models trained on graphemes and words.

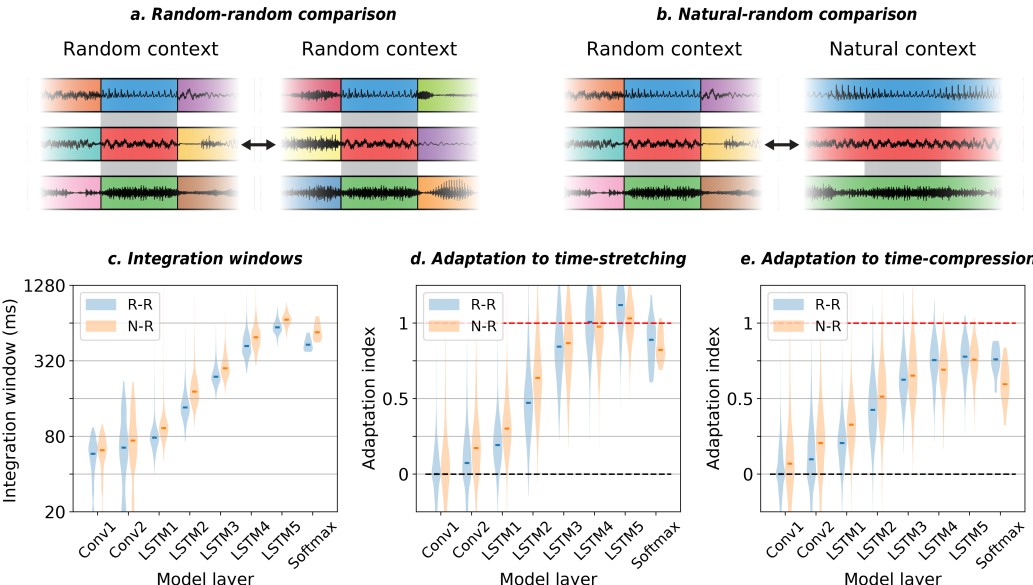

Appendix Figure 7: Comparing natural and random contexts. (a) Our paradigm allows us to consider two types of context: (1) random context, where a segment is surrounded by random other segments, and (2) natural context, where a segment is a subset of a longer segment and thus surrounded by its natural context in speech. One of the two contexts being compared must be random so that the two contexts differ, but the other context can be random or natural. Main text figures show results for random-random comparisons. This figure shows results comparing random-random (R-R) and natural-random (N-R) contexts. (b) Integration windows measured using natural speech for random-random and natural-random comparisons. Same format as Figure 3. (c-d) Adaptation indices from different model layers, reflecting the degree to which the unit integration windows are yoked to structure vs. time. Results are plotted separately for time-stretched (c) and time-compressed speech (d) and for random-random (blue) and natural-random (orange) comparisons. Same format as Figure 4. Similar trends are observed for random-random and natural-random comparisons.

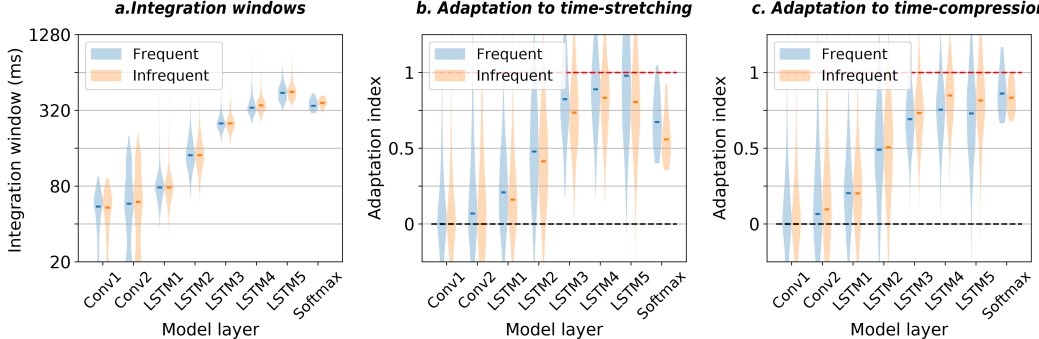

Appendix Figure 8: Results for frequent vs. infrequent words. We repeated all of our analyses using segments centered on either frequent or infrequent words. Specifically, we split the 1,095 unique words in our stimulus set in half based on their frequency of occurrence across the entire LibriSpeech corpus. We limited our analysis to words with average durations between 300 and 450 milliseconds, so that the average duration of the frequent and infrequent words was similar (368 ms for frequent and 384 ms for infrequent; without this constraint, the frequent words would have been considerably shorter than the infrequent words). This led to 336 and 375 unique words in the frequent and infrequent sets, respectively. The frequent words were 7 times more likely to occur in the corpus on average than the infrequent words. (a) Integration windows measured for frequent (blue) and infrequent (orange) words. (b-c) Adaptation indices for stretched (b) and compressed (c) speech, measured separately for frequent (blue) and infrequent (orange) words. Results are similar for the two sets of words. Horizontal lines in the violin plots indicate distribution medians.