# OpenReview forum: "Understanding Adaptive, Multiscale Temporal Integration In Deep Speech Recognition Systems"
_NeurIPS.cc/2021/Conference — NeurIPS 2021 Poster_

### Official Review · Reviewer_XEMX · 2021-07-15

**Rating:** 7
**Confidence:** 4

**Summary:**

This paper addressed the topic of identifying the integration window of hidden layers within ASRs. Specifically, the work applies temporal context invariance proposed in “Multiscale integration organizes hierarchical computation in human auditory cortex“ to study the integration window of each layer in ASRs.
The integration window is defined as the latency for which neurons provide saturated responses to a certain signal. In other words, the time a unit needs for identifying the underlying temporal signals is its integration window.
The approach infers integration window through comparing the correlation between the response to the same signal segment within different utterances, where the window length is the time when the correlation exceeds a predefined threshold (0.75 in the paper).
The paper also studies the effect of stretching and shrinking the audio, and showed that upper layers’ integration windows are more proportional to time scaling while lower layers are more invariant.
The time-variance study provides an insight on whether the approach is effective on inferring the integration window of each layer: if the approach or the underlying unit simply reflect the time scale of the signal then it should scale proportionally with the input, while if the approach or the unit only reflect a fixed time response then they should be completely invariance.
The time-variance study shows that all layers within an untrained ASR are invariant to the time scaling, where the trained ASR exhibits different variance at different layers.

**Limitations And Societal Impact:**

Authors have reasonably discussed their limitations.
Authors mentioned exploring how architecture can affect the analysis as a future work; it will be interesting to include self-attention-based components like Transformer/Conformer and compare how they differ from RNNs, in particular that Transformer/Conformer allows limiting the self-attention window and thus can exhibit distinct behaviors than RNNs.

**Main Review:**

This paper provides a mechanism to help understand the temporal window of underlying layers of ASRs, which allows gaining insights about ASRs’ behaviors and are quite meaningful to the speech community. The time-variance study further reveals each layer’s temporal behavior and proves the effectiveness of the approach. One small drawback is that the criterion for preparing the input signals makes it less flexible and harder to scale. On the other hand, given the access to the semantic of the input signal, a further analysis on how semantics affect integration windows can provide deeper understanding on the ASR behavior.
Related works for identifying the width of the receptive field in artificial neural networks should be discussed, for example like “Understanding the Effective Receptive Field in Deep Convolutional Neural Networks”.

**Time Spent Reviewing:**

4

---

> ### Author Response · Authors · 2021-08-10
> **Point-by-point response to XEMX**
>
> Thank you for your positive and constructive comments. We have done our best to respond to your specific feedback.
>
> ***In response to: “One small drawback is that the criterion for preparing the input signals makes it less flexible and harder to scale.”***
>
> The particular approach we used to select segments that surround words is not critical. The analysis can be performed with randomly selected segments and we have generally found similar results using both approaches. We will repeat our analyses using random segments to verify this. We used the approach that we did because we wanted the speech stimuli to be spoken at a moderate rate such that stretching and compression would not put the stimuli outside of their typical distribution.
>
> The choice of which segments to input to the analysis is of course an important one and allows the user to explore how the system operates in different regimes. For example, here we used this flexibility to examine the degree of adaptation for stretched and compressed stimuli.
>
> In general, we consider the simplicity and flexibility of the TCI method one of its primary strengths. The method does not require any knowledge about the response being measured. For example, the method can be used to measure the integration windows of biological neural responses, where the stimulus-response mapping is unknown and it is impossible to measure gradients. All that is needed are responses to a set of concatenated stimulus segments. The method of course benefits from having a large number of segments but the analysis can be batched by performing the analysis on subsets of segments and then averaging the correlation across batches, a point we will clarify in the revision. One limitation is that we can only measure the aggregate integration window across a large number of segments, unlike gradient-based methods that can measure the effective receptive field for a single stimulus, a point we will also clarify. If you have a specific concern about flexibility or scalability, we would be happy to address it in our revision.
>
> ***In response to: “On the other hand, given the access to the semantic of the input signal, a further analysis on how semantics affect integration windows can provide deeper understanding on the ASR behavior.”***
>
> Absolutely. For example, reviewer mPjw wondered whether integration windows might differ for frequent vs. uncommon words, an analysis we plan to perform and include in our revision. Future work could also compare integration windows for words that are more or less predictable given prior context or which are easier or harder to recognize since more difficult words might benefit from greater context.
>
> ***In response to: “Related works for identifying the width of the receptive field in artificial neural networks should be discussed, for example like “Understanding the Effective Receptive Field in Deep Convolutional Neural Networks”.”***
>
> We will expand our discussion of related methods for estimating receptive fields. The paper mentioned above uses the gradient of a unit with respect to the input to estimate the receptive field. Reviewer Dxi4 also noted that there are a variety of gradient-based methods for computing receptive fields or “salience maps”. Below, we have reproduced our response to Dxi4, which discusses the relationship between these gradient methods and our TCI method:
>
> The gradient of a function with respect to the input reflects how an infinitesimal change to the input alters that function. By contrast, the TCI method measures how large-scale stimulus changes, created by swapping one natural stimulus segment for another, alters the response. For a nonlinear system, there is no simple relationship between local and global changes and the two measures provide complementary information. As network depth increases, it is not uncommon for the gradient to increasingly resemble white noise (Balduzzi et al., 2017; Samek et al., 2021), which is likely related to the phenomena of adversarial examples, whereby a nearby input can produce dramatically different outputs (Goodfellow et al., 2014). As a consequence, the relationship between the local gradient and the overall computation performed by a highly nonlinear system is not always clear (Sundararajan et al., 2017).
>
> A variety of alternative visualization methods have been developed, many of which can be conceptualized as multiplying the gradient by the input (Bach et al., 2015; Selvaraju et al., 2016; Shrikumar et al., 2016, 2017; Ancona et al., 2017; Samek et al., 2021) and potentially integrating this quantity over several nearby inputs (Sundararajan et al., 2017). However, because the gradient is multiplied by the input, the result often reflects the input as much as the actual computation performed by the network (Ancona et al., 2017; Adebayo et al., 2018). Moreover, many of these methods are tailored to specific architectures (Springenberg et al., 2014; Selvaraju et al., 2016; Arras et al., 2017), complicating comparisons across models.
>
> For all of these reasons, it is useful to have a simple way of estimating the overall integration window across a stimulus set of interest, which we can define as the smallest time window within which stimuli alter the neural response and outside of which stimuli have little effect. Our paradigm is a relatively direct measure of this quantity, since we explicitly measure how invariant the response is across a variety of segment durations.
>
> Practically, the TCI method is simpler and more flexible than many existing methods. The method does not require gradients to be available; all that is needed is samples of the response itself. We can therefore use the TCI method to study a response where the stimulus-response mapping is unknown (for example in biological systems) or where gradients are difficult to compute. Our method is also agnostic of the input representation. As a consequence, we could train a model from a waveform representation or a spectrogram representation, and then apply our same analysis to compare the integration windows across these two models, which would be difficult if one were using gradient-based methods.
>
> Another common approach is to occlude (e.g. set to zero) pixels or patches of an input and to measure the magnitude of the change (Zeiler and Fergus, 2014; Zhou et al., 2014; Samek et al., 2021). This approach, while simple, often depends upon the size of the patch occluded (Ancona et al., 2017), only measures sensitivity to one type of change (occlusion), and is relatively expensive, since for a single stimulus, one typically measures the response to a large number of occluded inputs, and the whole analysis then needs to be repeated for a large number of stimuli to assess the overall receptive field. The TCI method provides a simple and comparatively efficient method for estimating the integration window across a large set of natural stimulus segments, without the need for artificial manipulations like occlusion.
>
> Of course, gradient-based and occlusion methods also have unique benefits in that they can more clearly indicate how the fine details of a particular stimulus (e.g., particular spectrotemporal bins) influence a response. We will clarify these different benefits in our revised paper.
>
> Adebayo J, Gilmer J, Muelly M, Goodfellow I, Hardt M, Kim B (2018) Sanity checks for saliency maps. arXiv preprint arXiv:181003292.
>
> Ancona M, Ceolini E, Öztireli C, Gross M (2017) Towards better understanding of gradient-based attribution methods for deep neural networks. arXiv preprint arXiv:171106104.
>
> Arras L, Montavon G, Müller K-R, Samek W (2017) Explaining recurrent neural network predictions in sentiment analysis. arXiv preprint arXiv:170607206.
>
> Bach S, Binder A, Montavon G, Klauschen F, Müller K-R, Samek W (2015) On pixel-wise explanations for non-linear classifier decisions by layer-wise relevance propagation. PloS one 10:e0130140.
>
> Balduzzi D, Frean M, Leary L, Lewis JP, Ma KW-D, McWilliams B (2017) The shattered gradients problem: If resnets are the answer, then what is the question? In: International Conference on Machine Learning, pp 342–350. PMLR.
>
> Goodfellow IJ, Shlens J, Szegedy C (2014) Explaining and harnessing adversarial examples. arXiv preprint arXiv:14126572.
>
> Samek W, Montavon G, Lapuschkin S, Anders CJ, Müller K-R (2021) Explaining deep neural networks and beyond: A review of methods and applications. Proceedings of the IEEE 109:247–278.
>
> Selvaraju RR, Das A, Vedantam R, Cogswell M, Parikh D, Batra D (2016) Grad-CAM: Why did you say that? arXiv preprint arXiv:161107450.
>
> Shrikumar A, Greenside P, Kundaje A (2017) Learning important features through propagating activation differences. In: International Conference on Machine Learning, pp 3145–3153. PMLR.
>
> Shrikumar A, Greenside P, Shcherbina A, Kundaje A (2016) Not just a black box: Learning important features through propagating activation differences. arXiv preprint arXiv:160501713.
>
> Springenberg JT, Dosovitskiy A, Brox T, Riedmiller M (2014) Striving for simplicity: The all convolutional net. arXiv preprint arXiv:14126806.
>
> Sundararajan M, Taly A, Yan Q (2017) Axiomatic attribution for deep networks. In: International Conference on Machine Learning, pp 3319–3328. PMLR.
>
> Zeiler MD, Fergus R (2014) Visualizing and understanding convolutional networks. In: European conference on computer vision, pp 818–833. Springer.
>
> Zhou B, Khosla A, Lapedriza A, Oliva A, Torralba A (2014) Object detectors emerge in deep scene cnns. arXiv preprint arXiv:14126856.

---

### Official Review · Reviewer_GCtL · 2021-07-16

**Rating:** 6
**Confidence:** 4

**Summary:**

The authors investigate the DeepSpeech2 speech recognition model using temporal context invariance (TCI), which was recently introduced and used to study temporal integration in biological systems. The analysis shows that the "integration windows" for which the intermediate outputs of DeepSpeech2 are approximately independent of additional context are organized hierarchically, with lower layers integrating across shorter, fixed, time windows, and progressively higher layers integrating over longer timescales, which they show are progressively more governed by the structural evolution of speech rather than absolute time.


**Ethical Concerns:**

No.

**Limitations And Societal Impact:**

Yes.

**Main Review:**

Strengths:

- Solid analysis of the temporal response properties of a well known speech recognition model.

Limitations:
- While interesting, the results are as expected, making them perhaps less significant. Also, DeepSpeech2 is an older speech model, and the findings fall short of identifying shortcomings or limitations of the model, or deep learning more broadly, to inform future research.
- The TCI analysis is applied only to one model, leaving contrastive analysis across architectures and applications for future work.
- I would have really liked to see the authors open source a TCI analysis framework for ML researchers to analyze their models with...

Current Assessment:

- A solid analysis of the DeepSpeech2 model with the TCI framework. Lower significance.

Post-rebuttal:

Thank you to the authors for their detailed responses. Based on the responses to the reviewers' questions regarding the analysis, and the clarification regarding the release of a TCI analysis tool, I have increased my score by 1 to 6. The rebuttal includes several promissory updates to the paper, including clearer algorithmic descriptions, and a contrastive analysis across architectures, which I expect will further strengthen the paper (the expected result being a "good" paper (7)).

**Time Spent Reviewing:**

3

---

> ### Author Response · Authors · 2021-08-10
> **Point-by-point response to GCtL**
>
> Thank you for your comments and critiques. We have addressed each below.
>
> ***In response to: “While interesting, the results are as expected, making them perhaps less significant. Also, DeepSpeech2 is an older speech model, and the findings fall short of identifying shortcomings or limitations of the model, or deep learning more broadly, to inform future research. The TCI analysis is applied only to one model, leaving contrastive analysis across architectures and applications for future work.”***
>
> We agree this is a limitation of our work. To address this comment as well as a related comment by mPjw, we are planning to train a model using purely feedforward (convolutional) and purely recurrent layers to test if both types of architectures are capable of learning adaptive, hierarchical integration windows. We expect that the recurrent layers might show greater adaptation as well as better performance. To control for performance, we will checkpoint the recurrent model so that we can select a model during training whose performance matches that of the purely feedforward model. For this analysis, we will use layers that are as similar as possible to those of the original model by simply replicating the convolutional and LSTM layers. If time permits, we will also train and analyze more modern feedforward architectures that include residual connections and self-attention.
>
> The goal of our work was to provide a toolkit for studying temporal integration in black-box machine learning models and to use this toolkit to understand how a popular ASR model integrates across time. We hope that this toolkit will help ML researchers better understand and ultimately improve their models, though this was not the explicit focus of our work. We note that integration windows / receptive fields are widely thought to be central to DNN performance and the effective integration window of a unit or layer is often difficult to determine from the architecture alone and can change substantially with training, as we show. We also suspect that the degree of adaptation will depend somewhat on the architecture, which we hope to show with our architectural analyses, and this insight maybe be useful in guiding future model development, particularly for tasks like speech recognition and language understanding where the structures of interests have highly variable durations.
>
> ***In response to: “I would have really liked to see the authors open source a TCI analysis framework for ML researchers to analyze their models with...”***
>
> Absolutely. We will release an open-source code framework for performing the TCI analysis described in this paper. The interface will work as follows:
>
> The user will specify a corpus of interest (e.g., a directory of sound files) and a range of timescales (e.g., from 50 milliseconds to 2 seconds). A module will then generate a set of TCI sequences by segmenting and concatenating the stimuli. The user will then run these stimuli through the model to generate model responses, and a second module will use these responses to calculate the cross-context correlation and estimate the integration window.

---

### Official Review · Reviewer_Dxi4 · 2021-07-16

**Rating:** 7
**Confidence:** 4

**Summary:**

The paper proposes an method of analyzing individual neurons in a neural network based on monitoring the output of a particular neuron while changing the size of input segments at the center and their context. The concept of integration window is introduced and defined as the size of the center segment that achieves a maximum correlation of 0.75 while changing the context. The paper uses DeepSpeech2 as a model to demonstrate the approach, and finds that the integration window significantly increases as we move deeper into the network and further away from the input. Further experiments are done to study the impact of stretching and compression the input spectrogram on the integration windows. The results show that the integration windows "adapt" better at the higher layers, meaning that stretching the input also stretches the integration windows, while the lower layers adapt less.

**Limitations And Societal Impact:**

The authors have addressed the limitation of the approach and its potential social impact.

**Main Review:**

I am giving the paper a score 7, because the approach is novel and the results of the analyzing DeepSpeech2 themselves are a significant contribution.

# Novelty

The approach is novel, though there are a few closely related concepts not mentioned in the prior work. Changing the context while measuring the output in the limit is in fact measuring the gradient. A few gradient-based approach are listed below.

Deep inside convolutional networks: visualising image classification models
Karen Simonyan, Andrea Vedaldi, Andrew Zisserman
Workshop at ICLR, 2014

Striving for simplicity: the all convolution
Jost Tobias Springenberg, Alexey Dosovitskiy, Thomas Brox, Martin Riedmiller
arXiv:1412.6806

Grad-CAM: Why did you say that?
Ramprasaath R. Selvaraju, Abhishek Das, Ramakrishna Vedantam, Michael Cogswell, Devi Parikh, Dhruv Batra
ICCV, 2017


# Approach

The major assumption of the approach is that analyzing the response of segments with arbitrary context concatenated is meaningful. The network is trained on continuous speech without the artifacts of concatenation, so in principle the concatenated input segments are considered out of domain and the results could really be anything. However, from the experiments, it seems that we have enough evidence to believe that the results are reliable.

The same applies to the stretching and compression experiments. I believe the input to the network is in the log magnitude domain, and a simple interpolation might introduce artifacts that are simply not natural and can be considered out of domain. Again, the experiments seem to suggest that the results are reliable.

Another relative strict constraint not considered in the approach is that fact that first two convolution layers have finite receptive fields. In other words, the integration windows are capped by the hard constraints.

As mentioned before, the approach is reminiscent to the gradient-based approaches. It would be great to use another set of analysis approaches to at least see if the findings are consistent.


# Presentation

The paper is a nice read.

I find the caption in Figure 2 a bit too long, and some of them are actually repeated in the main text.

I also find the text description in 3.3 extremely hard to follow. This is a good place to use mathematical notations to precisely define what is being computed to improve understanding.

**Time Spent Reviewing:**

6

---

> ### Author Response · Authors · 2021-08-10
> **Point-by-point response to Dxi4**
>
> Thank you for these comments. We have described below how we will address each of them.
>
> ***In response to: “The approach is novel, though there are a few closely related concepts not mentioned in the prior work. Changing the context while measuring the output in the limit is in fact measuring the gradient. A few gradient-based approach are listed below...”***
>
> We will revise the paper to describe these related methods and their relationship to the TCI paradigm. Reviewer XEMX had a similar comment.
>
> The gradient of a function with respect to the input reflects how an infinitesimal change to the input alters that function. By contrast, the TCI method measures how large-scale stimulus changes, created by swapping one natural stimulus segment for another, alters the response. For a nonlinear system, there is no simple relationship between local and global changes and the two measures provide complementary information. As network depth increases, it is not uncommon for the gradient to increasingly resemble white noise (Balduzzi et al., 2017; Samek et al., 2021), which is likely related to the phenomena of adversarial examples, whereby a nearby input can produce dramatically different outputs (Goodfellow et al., 2014). As a consequence, the relationship between the local gradient and the overall computation performed by a highly nonlinear system is not always clear (Sundararajan et al., 2017).
>
> A variety of alternative visualization methods have been developed, many of which can be conceptualized as multiplying the gradient by the input (Bach et al., 2015; Selvaraju et al., 2016; Shrikumar et al., 2016, 2017; Ancona et al., 2017; Samek et al., 2021) and potentially integrating this quantity over several nearby inputs (Sundararajan et al., 2017). However, because the gradient is multiplied by the input, the result often reflects the input as much as the actual computation performed by the network (Ancona et al., 2017; Adebayo et al., 2018). Moreover, many of these methods are tailored to specific architectures (Springenberg et al., 2014; Selvaraju et al., 2016; Arras et al., 2017), complicating comparisons across models.
>
> For all of these reasons, it is useful to have a simple way of estimating the overall integration window across a stimulus set of interest, which we can define as the smallest time window within which stimuli alter the neural response and outside of which stimuli have little effect. Our paradigm is a relatively direct measure of this quantity, since we explicitly measure how invariant the response is across a variety of segment durations.
>
> Practically, the TCI method is simpler and more flexible than many existing methods. The method does not require gradients to be available; all that is needed is samples of the response itself. We can therefore use the TCI method to study a response where the stimulus-response mapping is unknown (for example in biological systems) or where gradients are difficult to compute. Our method is also agnostic of the input representation. As a consequence, we could train a model from a waveform representation or a spectrogram representation, and then apply our same analysis to compare the integration windows across these two models, which would be difficult if one were using gradient-based methods.
>
> Of course, gradient-based visualization methods also have unique benefits in that they can more clearly indicate how the fine details of a particular stimulus (e.g., particular spectrotemporal bins) influence a response. We will clarify these different benefits in our revised paper.
>
> Adebayo J, Gilmer J, Muelly M, Goodfellow I, Hardt M, Kim B (2018) Sanity checks for saliency maps. arXiv preprint arXiv:181003292.
>
> Ancona M, Ceolini E, Öztireli C, Gross M (2017) Towards better understanding of gradient-based attribution methods for deep neural networks. arXiv preprint arXiv:171106104.
>
> Arras L, Montavon G, Müller K-R, Samek W (2017) Explaining recurrent neural network predictions in sentiment analysis. arXiv preprint arXiv:170607206.
>
> Bach S, Binder A, Montavon G, Klauschen F, Müller K-R, Samek W (2015) On pixel-wise explanations for non-linear classifier decisions by layer-wise relevance propagation. PloS one 10:e0130140.
>
> Balduzzi D, Frean M, Leary L, Lewis JP, Ma KW-D, McWilliams B (2017) The shattered gradients problem: If resnets are the answer, then what is the question? In: International Conference on Machine Learning, pp 342–350. PMLR.
>
> Goodfellow IJ, Shlens J, Szegedy C (2014) Explaining and harnessing adversarial examples. arXiv preprint arXiv:14126572.
>
> Samek W, Montavon G, Lapuschkin S, Anders CJ, Müller K-R (2021) Explaining deep neural networks and beyond: A review of methods and applications. Proceedings of the IEEE 109:247–278.
>
> Selvaraju RR, Das A, Vedantam R, Cogswell M, Parikh D, Batra D (2016) Grad-CAM: Why did you say that? arXiv preprint arXiv:161107450.
>
> Shrikumar A, Greenside P, Kundaje A (2017) Learning important features through propagating activation differences. In: International Conference on Machine Learning, pp 3145–3153. PMLR.
>
> Shrikumar A, Greenside P, Shcherbina A, Kundaje A (2016) Not just a black box: Learning important features through propagating activation differences. arXiv preprint arXiv:160501713.
>
> Springenberg JT, Dosovitskiy A, Brox T, Riedmiller M (2014) Striving for simplicity: The all convolutional net. arXiv preprint arXiv:14126806.
>
> Sundararajan M, Taly A, Yan Q (2017) Axiomatic attribution for deep networks. In: International Conference on Machine Learning, pp 3319–3328. PMLR.
>
> ***In response to: “The major assumption of the approach is that analyzing the response of segments with arbitrary context concatenated is meaningful. The network is trained on continuous speech without the artifacts of concatenation, so in principle the concatenated input segments are considered out of domain and the results could really be anything. However, from the experiments, it seems that we have enough evidence to believe that the results are reliable.”***
>
> This is an excellent point. Our TCI method in fact allows us to consider two types of context. The first type of context is the one discussed in our submission, where a segment is surrounded by random other segments. The second type of context is when a segment is a subset of a longer segment and thus is surrounded by its natural context. Since our analysis requires that the two contexts differ, one context has to be random, but the other can be random or natural. This difference is illustrated in this figure: <https://i.ibb.co/L08mvvX/natural-random-context.png>.
>
> The results described in our submission were all based on analyses where both contexts were random. However, in previous analyses, we have observed very similar results when comparing random and natural contexts. We are currently re-running all of our analyses using this alternative comparison and will include a Supplemental Figure that describes the results.
>
> In our revision, we will clarify all of these points. We will indicate that our paradigm does rely on random concatenation, which is unnatural, but also allows us to compare random and natural contexts. Finally, we will repeat our analyses using windowing at the segment boundary to minimize the most glaring boundary artifacts.
>
> ***In response to: “The same applies to the stretching and compression experiments. I believe the input to the network is in the log magnitude domain, and a simple interpolation might introduce artifacts that are simply not natural and can be considered out of domain. Again, the experiments seem to suggest that the results are reliable.”***
>
> We agree. We have performed these same analyses using the time stretching/compression algorithm built into the SoX software package, observing similar results. We will check that our results hold using a few other algorithms that rely on both time-based overlap-and-add methods as well as phase vocoding, which have different benefits and weaknesses. All of these algorithms uniformly stretch / compress the stimulus, which is clearly unnatural. To address this issue, we will repeat our analyses using smaller and larger amounts of stretching / compression, since the degree of distortion is presumably related to the magnitude of stretching / compression. It is possible that network adaptation might be slightly larger for smaller amounts of stretching / compression due to greater naturalness.
>
> ***In response to: “Another relative strict constraint not considered in the approach is that fact that first two convolution layers have finite receptive fields. In other words, the integration windows are capped by the hard constraints.”***
>
> In practice, the effective integration windows estimated by our method are much shorter than the maximum integration window imposed by the architecture. Moreover, training causes the integration windows of the earlier layers to shrink substantially, which also cannot be accounted for by the architectural constraint. We will clarify both these points in the revision.
>
> As expected, the cross-context correlation does equal exactly 1 when the segment duration exceeds the maximum integration imposed by the architecture, indicating a fully invariant response. This differs from the recurrent layers, which never become completely context invariant. This effect is evident in Supplement Figure 2, which plots the maximum cross-context correlation across all segment durations for units from different model layers. We will highlight this point in the revised paper.

---

> ### Author Response · Authors · 2021-08-10
> **(Continued) Point-by-point response to Dxi4**
>
> ***In response to: “As mentioned before, the approach is reminiscent to the gradient-based approaches. It would be great to use another set of analysis approaches to at least see if the findings are consistent.”***
>
> If time permits, we will test if our findings can also be observed using the gradient of the function with respect to a spectrogram. We will measure the gradient magnitude for a large number of sound segments, average the gradient across frequency and sound segments, and calculate the width of the resulting receptive field. We will then repeat our analyses using this quantity as our measure of the integration window. Calculating gradients for recurrent layers is not totally trivial and many gradient-based methods require specialized analysis tools (Arras et al., 2017), so we cannot guarantee that we will be able to get this done by the revision deadline, given all of the other requested changes.
>
> ***In response to: “I find the caption in Figure 2 a bit too long, and some of them are actually repeated in the main text.”***
>
> We will shorten the caption and refer readers to the main text for additional details.
>
> ***In response to: “I also find the text description in 3.3 extremely hard to follow. This is a good place to use mathematical notations to precisely define what is being computed to improve understanding.”***
>
> We agree that our description could have been more precise, a comment also echoed by Reviewer mPjw. We will follow your suggestion and describe our method using mathematical notation.
>
> Here is a more precise description (this description is the same as that provided in our response to mPjw):
>
> Our key analysis involves correlating the response of a unit to different stimulus segments across two different contexts, an analysis we refer to as the “cross-context correlation”. The analysis is applied separately to different segment durations.
>
> For each segment duration, we have two stimulus sequences ($x_A[t]$ and $x_B[t]$). The stimulus sequences could be vector-valued as in a spectrogram or scalar as in a waveform. Each sequence contains all the segments from that duration, randomly ordered and concatenated. For simplicity, we will ignore the fact that we divided our sequences into 45-second stimuli due to memory limitations and assume that there are just two very long sequences, each of which creates a unique context for each segment.
>
> Each unit in our model produces an output sequence in response to these two stimuli (we do not specify the index of the unit to simplify notation):
>
> $$r_A[t] = f(x_A[t])$$
> $$r_B[t] = f(x_B[t])$$
>
> Where $f(\cdot)$ is a scalar function specifying the stimulus response mapping, here a deep network. From these two response sequences we create two segment-by-time matrices, which we refer to as the segment-aligned response (SAR) matrix (see Figure 2B for a schematic). Each SAR matrix is defined as follows:
>
> $$SAR_A[s,t] = r_A[t-o_A[s]]$$
> $$SAR_B[s,t] = r_B[t-o_B[s]]$$
>
> where $o_A[s]$ is the onset in samples of segment $s$ in sequence $A$, and $o_B[s]$ is the onset of that same segment in sequence $B$. Thus, each row of the SAR matrix contains the response sequence that surrounds a particular stimulus segment. Corresponding rows contain the response sequence to the same segment but surrounded by different context segments.
>
> The cross-context correlation ($\rho_{ccc}[t]$) is computed by correlating corresponding columns across the two SAR matrices:
>
> $$\rho_{ccc}[t] = \operatorname{corr}(SAR_A[:,t], SAR_B[:,t])$$
>
> We used the Pearson correlation, but see no reason why another measure of dependence couldn’t be swapped in.
>
> The set of time lags ($t$) relative to segment onset should be larger than the integration windows being measured. Here we used time lags from -1 seconds to $T$+1 seconds, where $T$ is the segment duration. The sampling rate was 100 Hz.

---

### Official Review · Reviewer_mPjw · 2021-07-17

**Rating:** 6
**Confidence:** 4

**Summary:**

In this work, the authors conduct analyses to understand how ASR models -- specifically, DeepSpeech2 -- integrates temporal information in various layers of the model. The analysis is based on the temporal context invariance (TCI) paradigm proposed for biological neural systems which is based on studying the output from a neuron from segments of different lengths in two different contexts; correlated outputs indicate sensitivity to the particular input and the length of the input segments indicate how long the segments need to be to produce relatively invariant outputs. The analyses indicate that the networks become increasingly sensitive to longer time scales and that earlier layers in the network are more responsive to static time windows whereas deeper layers are sensitive to the duration of the structures to which it responds.

**Limitations And Societal Impact:**

Apart from the limitations mentioned above, I do not see any negative societal impacts of this work.

**Main Review:**

Overall, I found this paper to be really interesting and intriguing. Generally speaking, the paper is excellently presented. The work is extremely well motivated, with detailed discussion of related work. The experiment in Section 4 was particularly interesting. However, I do have a number of comments that I would like the authors to address:

- **Dataset creation and SAR Matrix Description**: While I generally understood the gist of the argument, one aspect of the paper that I found hard to follow, was the fact that the paper presents all ideas in prose rather than through mathematical notation which makes the work somewhat imprecise and ambiguous. In particular, Section 3.3 which describes the creation of the SAR matrix is somewhat vague (e.g., “. First, we organize the response timecourses to all segments of a given duration as a matrix, which we refer to as the segment-aligned response (SAR) matrix (Figure 2b). Each row of the SAR matrix contains the response timecourse surrounding a single segment, aligned to segment onset. Different rows thus correspond to different segments and different columns correspond to different lags relative to segment onset.”) I wasn’t sure what the authors meant by “timecourses”. I think this section would be much clearer and easier to understand if this was described with precise mathematical notation. Currently, this is described in Section 3.3 and the caption to Figure 2 which together take up almost 2 pages of the paper.

- **Specific choice of stimuli**: I think that Section 3.1 which describes how the dataset used to build the SAR matrix was constructed could be re-written to be a bit more precise. In particular, something I was curious about was what fraction of the words chosen for the analysis involve common versus uncommon words. For example, I could imagine that the behavior is very different for common words such as “a”, “an”, “the”, etc. versus more uncommon words. I would be curious to see some analysis along these lines.

- **Bidirectional Recurrent Layers**: In the caption to figure 2, the authors state: “If the segment duration is longer than the integration window (top panel), there will be a moment when the window is fully contained within each segment. As a consequence, the response at that moment will by definition be unaffected by the surrounding context segments.” Given that the model uses bidirectional recurrent layers, is it not still possible for the model to be affected by surrounding context segments? Could the authors please clarify if this is an assumption in the TCI paradigm, or if this still holds in some way.

- **Limitations**: A couple of limitations of this work is that the analyses are all limited to a particular model architecture. It would be interesting to conduct similar analyses on a variety of different architectures (e.g., only LSTMs, only Feedforward layers, etc.), and to vary the output units (e.g., sentence-pieces/words instead of graphemes) to see if the observations are similar different in these cases.


**Time Spent Reviewing:**

5

---

> ### Author Response · Authors · 2021-08-10
> **Point-by-point response to mPjw**
>
> Thank you for your thoughtful comments and critiques. Below we indicate how we will address each of them.
>
> ***In response to: “Dataset creation and SAR Matrix Description: While I generally understood the gist of the argument, one aspect of the paper that I found hard to follow, was the fact that the paper presents all ideas in prose rather than through mathematical notation which makes the work somewhat imprecise and ambiguous. In particular, Section 3.3 which describes the creation of the SAR matrix is somewhat vague (e.g., “. First, we organize the response timecourses to all segments of a given duration as a matrix, which we refer to as the segment-aligned response (SAR) matrix (Figure 2b). Each row of the SAR matrix contains the response timecourse surrounding a single segment, aligned to segment onset. Different rows thus correspond to different segments and different columns correspond to different lags relative to segment onset.”) I wasn’t sure what the authors meant by “timecourses”. I think this section would be much clearer and easier to understand if this was described with precise mathematical notation. Currently, this is described in Section 3.3 and the caption to Figure 2 which together take up almost 2 pages of the paper.”***
>
> We agree our description could have been more precise, a comment also echoed by reviewer Dxi4. We will follow your suggestion and specify both the SAR matrix and the cross-context correlation using mathematical notation. We will also shorten the caption in Figure 2 and refer the readers to the main text for a complete description.
>
> Here is a more precise description of the method, which will be included in the revision (this description is the same as that provided in our response to Dxi4):
>
> Our key analysis involves correlating the response of a unit to different stimulus segments across two different contexts, an analysis we refer to as the “cross-context correlation”. The analysis is applied separately to different segment durations.
>
> For each segment duration, we have two stimulus sequences ($x_A[t]$ and $x_B[t]$). The stimulus sequences could be vector-valued as in a spectrogram or scalar as in a waveform. Each sequence contains all the segments from that duration, randomly ordered and concatenated. For simplicity, we will ignore the fact that we divided our sequences into 45-second stimuli due to memory limitations and assume that there are just two very long sequences, each of which creates a unique context for each segment.
>
> Each unit in our model produces an output sequence in response to these two stimuli (we do not specify the index of the unit to simplify notation):
>
> $$r_A[t] = f(x_A[t])$$
> $$r_B[t] = f(x_B[t])$$
>
> Where $f(\cdot)$ is a scalar function specifying the stimulus response mapping, here a deep network. From these two response sequences we create two segment-by-time matrices, which we refer to as the segment-aligned response (SAR) matrix (see Figure 2B for a schematic). Each SAR matrix is defined as follows:
>
> $$SAR_A[s,t] = r_A[t-o_A[s]]$$
> $$SAR_B[s,t] = r_B[t-o_B[s]]$$
>
> where $o_A[s]$ is the onset in samples of segment $s$ in sequence $A$, and $o_B[s]$ is the onset of that same segment in sequence $B$. Thus, each row of the SAR matrix contains the response sequence that surrounds a particular stimulus segment. Corresponding rows contain the response sequence to the same segment but surrounded by different context segments.
>
> The cross-context correlation ($\rho_{ccc}[t]$) is computed by correlating corresponding columns across the two SAR matrices:
>
> $$\rho_{ccc}[t] = \operatorname{corr}(SAR_A[:,t], SAR_B[:,t])$$
>
> We used the Pearson correlation, but see no reason why another measure of dependence couldn’t be swapped in.
>
> The set of time lags ($t$) relative to segment onset should be larger than the integration windows being measured. Here we used time lags from -1 seconds to $T$+1 seconds, where $T$ is the segment duration. The sampling rate was 100 Hz.
>
> ***In response to: “Specific choice of stimuli: I think that Section 3.1 which describes how the dataset used to build the SAR matrix was constructed could be re-written to be a bit more precise.”***
>
> Below is a more precise, algorithmic description. Please let us know if anything remains unclear.
>
> The input to our analysis is a set of sound segments. The duration of these segments varied between 20 milliseconds and 6.28 seconds in pseudo-logarithmic steps (20, 40, 60, 80, 120, 180, 240, 360, 500, 720, 1040, 1500, 2140, 3060, 4380, 6280 ms). These sound segments were excerpted from the LibriSpeech corpus. We attempted to choose segments that were composed of words spoken at a moderate rate such that stretching or compressing the word would not place the word outside of its typical duration. To this end, we first selected all words spoken at least 100 times in the corpus. For each word, we computed a histogram of durations for that word and randomly selected up to 50 of its utterances that fell within the central 10% of the distribution (45-55th percentile). This resulted in a pool of 28,280 word utterances across 1,451 unique words. From this pool, we randomly selected 10,000 utterances. There were 1,354 unique words across this pool of 10,000 utterances. Finally, for each utterance, we excerpted 16 segments, one per duration, with the middle of the word at the center of the segment (if the segment duration was shorter than the word, we excerpted just the middle portion of the word). This selection procedure resulted in 10,000 unique segments per segment duration.
>
> All of the segments from a given duration were randomly ordered and then concatenated. We used two different random ordering of the segments such that each segment occurred in two different contexts (Fig 2A). Due to limited memory, we divided the randomly ordered segments into 45-second stimuli, which were then input to the network. Each two consecutive 45-second stimuli, overlapped in 12.56 seconds such that each 45-second stimulus had 6.28 seconds of context on each side that was not used for analysis to avoid artifacts due to batching of the stimulus. The segments were concatenated in the spectrogram domain. The concatenated spectrogram sequence was input to the DeepSpeech2 network.
>
> ***In response to: “In particular, something I was curious about was what fraction of the words chosen for the analysis involve common versus uncommon words. For example, I could imagine that the behavior is very different for common words such as “a”, “an”, “the”, etc. versus more uncommon words. I would be curious to see some analysis along these lines.”***
>
> This is an interesting question. It is possible that integration windows might show greater adaptation for common compared with uncommon words. We will test this by repeating our analysis using only frequent and only infrequent words. For this analysis, we will attempt to match the duration of the common and uncommon words, since the common words are likely shorter in duration. Thank you for this suggestion.
>
> Here is a histogram of word frequency across all 1,354 unique words included in our analysis, showing that there is a wide distribution: <https://i.ibb.co/s2hhHb9/word-frequency-histogram.png>. The frequencies are calculated from the LibriSpeech corpus.
>
> ***In response to: “Bidirectional Recurrent Layers: In the caption to figure 2, the authors state: “If the segment duration is longer than the integration window (top panel), there will be a moment when the window is fully contained within each segment. As a consequence, the response at that moment will by definition be unaffected by the surrounding context segments.” Given that the model uses bidirectional recurrent layers, is it not still possible for the model to be affected by surrounding context segments? Could the authors please clarify if this is an assumption in the TCI paradigm, or if this still holds in some way.”***
>
> Our paper focuses on measuring the effective integration window of a model response, which often differs substantially from the maximum integration window given by the architecture. For a bidirectional RNN, the maximum integration window is potentially unbounded. However, our analysis shows that most units have a relatively short (up to a few hundred milliseconds) effective integration window, within which speech stimuli greatly alter the response and outside of which stimuli have little effect.
>
> At the same time, one can observe the consequence of the nearly unbounded nature of a bidirectional RNN in our data. In particular, the cross-context correlation for the RNN layers never exactly reaches 1 (though it gets close) even for very long segment durations, which is not true for our convolutional layers (see Supplement Figure 2 which plots the maximum cros-context correlation across all segment durations). This observation demonstrates that there is a relatively short compact integration window within which stimuli substantially alter the response as well as a long tail with much smaller effects.
>
> The significance of the RNN being bidirectional is that future stimuli can alter the response. As a consequence, the cross-context correlation starts to rise before the onset of the central segment, an effect that is visible in the second example unit shown in Figure 2c.
>
> We will clarify all of these points in our revision.

---

> ### Author Response · Authors · 2021-08-10
> **(Continued) Point-by-point response to mPjw**
>
> ***In response to: “Limitations: A couple of limitations of this work is that the analyses are all limited to a particular model architecture. It would be interesting to conduct similar analyses on a variety of different architectures (e.g., only LSTMs, only Feedforward layers, etc.), and to vary the output units (e.g., sentence-pieces/words instead of graphemes) to see if the observations are similar different in these cases.”***
>
> We agree that this is one of the biggest limitations of our current paper. We are currently training a model with purely feedforward (convolutional) and purely recurrent layers to test if both architectures are able to learn adaptive integration windows. For this analysis, we will try and hew as closely as possible to the original DeepSpeech2 model by replicating the convolutional and LSTM layers already present in the model. Based on some preliminary analyses and prior theoretical work, we suspect that a purely recurrent model might show greater adaptation. We also suspect that the recurrent model might show better performance and to control for this possibility, we will checkpoint the recurrent model during training and a select a model with equal performance to the feedforward model. We are confident that we will have time to train and analyze these models prior to the revision deadline. If time permits, we will also train and analyze more modern feedforward architectures that include residual connections and self-attention.
>
> We will also analyze a model that attempts to predict words labels instead of graphemes. We have the time-aligned word labels in hand for the LibriSpeech corpus and have no reason to believe this will be particularly difficult or time-consuming.

---

### Decision · Program_Chairs · 2021-09-27

**Decision:**

Accept (Poster)

**Comment:**

In this paper the authors use temporal context invariance (TCI) to understand temporal integration in black-box deep neural networks. The study is carried out using the DeepSpeech 2 model. The authors find that integratoin windows are specialized to different time scales in different layers,  the higher the layers the more sensitive to longer time scales.  In addition, lower layers are more sensitive to static windows while higher layers are more adaptive to duration of structures.  All reviewers find the work interesting and novel in methodology.  The hierarchical structure and pattern of response revealed in the experiments, despite in line with some previously observations/speculations, are still quite interesting and intriguing.  The method investigated in the paper also can potentially provide a new tool to the community to gain insights into the black-box machine learning models.  I would recommend accept.   The authors should address the questions raised by the reviewers in the revision. If possible, please also include new experimental results on other model architectures to make the paper stronger.